

# KIR gene content imputation from single-nucleotide polymorphisms in the Finnish population

Jarmo Ritari, Kati Hyvärinen, Jukka Partanen and Satu Koskela

Finnish Red Cross Blood Service, Helsinki, Finland

## ABSTRACT

The killer cell immunoglobulin-like receptor (KIR) gene cluster on chromosome 19 encodes cell surface glycoproteins that bind class I human leukocyte antigen (HLA) molecules as well as some other ligands. Through regulation of natural killer (NK) cell activity KIRs participate in tumour surveillance and clearing viral infections. KIR gene gene copy number variation associates with the outcome of transplantations and susceptibility to immune-mediated diseases. Inferring KIR gene content from genetic variant data is therefore desirable for immunogenetic analysis, particularly in the context of growing biobank genome data collections that rely on genotyping by microarray. Here we describe a stand-alone and freely available gene content imputation for 12 KIR genes. The models were trained using 807 Finnish biobank samples genotyped for 5900 KIR-region SNPs and analysed for KIR gene content with targeted sequencing. Cross-validation results demonstrate a high mean overall accuracy of 98.5% (95% CI [97.0–99.2]%) which compares favourably with previous methods including short-read sequencing based approaches.

# INTRODUCTION

Killer cell immunoglobulin-like receptors (KIRs) regulate the activity of natural killer (NK) cells and a subset of T cells *via* inhibitory and activating signals. Through their KIR molecules NK cells detect phenotypic change in a target cell. KIRs recognise human leukocyte antigen (HLA) class I molecules as cognate ligands, limiting to particular HLA allotypes within the serological HLA-C1 and C2 allele groups (*Wroblewski, Parham & Guethlein, 2019*) HLA-Bw4 motif, HLA-A3/11, HLA-G and HLA-F (*Garcia-Beltran et al., 2016*). The functional difference between inhibitory and activating KIRs is determined by the presence or absence of a cytoplasmic immunoreceptor tyrosine-based inhibitory (ITIM) protein motif, respectively. In the absence of constitutive signaling conveyed by an inhibitory KIR binding to its class I ligand, NK cell cytotoxic activity and cytokine production are triggered (*Lanier, 2008*).

According to the missing-self hypothesis, NK cells recognise tumour or virally infected cells that attempt to evade T cell mediated immunity by downregulating their cell surface HLA-molecules that present intracellular antigens to T cells. Activating KIRs, in contrast,

Corresponding authors
Jarmo Ritari, jarmo.ritari@gmail.com
Satu Koskela,
satu.koskela@veripalvelu.fi

are thought to recognise surface molecules indicative of aberrant host cell activity such as an exceptionally high surface density of HLA class I molecules even though in some cases the ligand remains unknown (*Ivarsson, Michaëlsson & Fauriat, 2014*). Activating KIRs have lower affinity to their ligands than inhibitory KIRs (*Stewart et al., 2005*) most likely owing to NK cell education to maintain self-tolerance. However, upon receiving a sufficient positive stimulus, they are able induce NK cell activation and target cell lysis. A vast majority of genetic associations of KIRs with cancer, autoimmunity and infectious diseases are attributed to variation in activating KIRs (*Parham & Guethlein, 2018*).

The KIR gene cluster on the human chromosome 19q13.4 encodes fifteen relatively homologous KIR genes and two pseudogenes, constituting two main haplotypes: A and B (https://www.ebi.ac.uk/ipd/kir/sequenced_haplotypes.html). The group A haplotype consists of *KIR2DL3, KIR2DL1, KIR2DL4, KIR3DL1* and *KIR2DS4* genes, of which all except KIR2DS4 are inhibitory. The group B haplotype, on the other hand, is more diverse being characterised by the presence of at least one of *KIR2DS2, KIR2DL2, KIR2DL5, KIR2DS5, KIR3DS1, KIR2DS3* or *KIR2DS1* genes (*Bashirova et al., 2006*). Thus, the group B haplotype harbours several genes, whereas the only activating receptor, *KIR2DS4,* of the group A is in a significant proportion of Caucasians a non-functional truncated variant (*Maxwell et al., 2002*; *Bontadini et al., 2006*), rendering about 40% of group A homozygotes solely inhibitory. Approximately 55% of haplotypes are mixtures between group A and B (*Middleton & Gonzelez, 2010*), making the haplotype structure highly variable in the population. Allelic diversity within KIRs is equally high with at least a few hundred known polymorphisms (https://www.ebi.ac.uk/ipd/kir/stats.html), which can affect class I ligand affinity (*Carr, Pando & Parham, 2005*; *Frazier et al., 2013*).

Discovery and interpretation of KIR gene and haplotype associations in large biobank genome data collections can be facilitated by imputation of KIR content from single-nucleotide polymorphisms (SNPs) genotyped by microarray. Furthermore, in organ or stem cell transplantation setting the KIR locus offers additional genetic information for donor selection and prediction of clinical outcome (*Cooley et al., 2010*; *Impola et al., 2014*; *Littera et al., 2017*), and for many of these clinical genome datasets SNP microarray provides the most cost-effective genotyping platform as well. To date, several KIR copy number or gene content analysis methods have been implemented for sequencing data (*Norman et al., 2016*; *Maniangou et al., 2017*; *Wagner et al., 2018*; *Chen et al., 2020*; *Roe & Kuang, 2020*), but to our knowledge only one SNP-based approach exists so far (*Vukcevic et al., 2015*). These approaches reach a high accuracy which makes KIR inference reliable enough for research and even practical clinical applications. However, regarding biobank data, a stand-alone application that does not require submitting individual genotype data to external servers would be essential. To this end, we have implemented a random forest (RF) based KIR gene content prediction in the R environment exploiting SNP data. The reference data used for model fitting comprises KIR genotypes determined by targeted sequencing and 5774 genotyped SNPs in the KIR chromosomal region. Based on prediction of an independent subset of data, our results demonstrate a mean overall accuracy of 99.2% which is comparable to previously published methods.

## MATERIALS AND METHODS

### Subjects

Blood donor genomic DNA and genotype data were obtained from the Blood Service Biobank, Helsinki, Finland. The samples were collected from Finnish blood donors who had given a written informed consent for biobank research according to the Finnish Biobank Act (688/2012) supervised by the Finnish Medicines Agency (Fimea). The use of the biobank data in this study was approved by the Blood Service Biobank (002-2018).

### Genotyping

Genotyping of samples was originally performed on a customized ThermoFisher Axiom array at the Thermo Fisher genotyping service facility (San Diego, USA) as a part of the FinnGen project. After the embargo period, imputed genotypes were returned to the Blood Service Biobank. Genotypes from altogether 807 samples with KIR type data were available for this study.

Genotype calling and quality control steps are described in https://finngen.gitbook.io/documentation/methods/genotype-imputation. The array marker files can be downloaded from https://www.finngen.fi/en/researchers/genotyping. The protocol for genotype data liftover to hg38/GRCh38 is described in detail in https://www.protocols.io/view/genotyping-chip-data-lift-over-to-reference-genome-xbhfij6?version_warning=no, and genotype imputation protocol is described in https://www.protocols.io/view/genotype-imputation-workflow-v3-0-xbgfijw. The FinnGen array was designed to provide an imputation grid to cover as much of the genetic variation as possible. The SNP content in the KIR region (chr19 54.4–55.1 Mb, GRCh38/hg38) used in this study was generated by the FinnGen imputation and quality control pipeline described above.

KIR genotyping at absence-presence level of 818 samples was purchased from Histogenetics LLC (NY, USA) in two parts. The first randomly selected set of 473 biobank samples was complemented with another set of 345 samples to ensure that each KIR gene and both haplotype groups occurred at high enough frequencies in the reference data. Altogether 16 KIR genes were genotyped: 2DL1, 2DL2, 2DL3, 2DL5, 2DP1, 2DS1, 2DS2, 2DS3, 2DS4, 2DS5, 3DL1, 3DS1, 2DL4, 3DL2, 3DL3, and 3DP1.

### Imputation models

The SNP genotype and KIR typing data were combined into a single data set consisting of 807 KIR typed and SNP genotyped samples. SNP variants in the KIR region with missing data and KIR genes that showed zero variance were excluded from analysis. Based on the latter criterion, the KIR framework genes 2DL4, 3DL2, 3DL3, and 3DP1 were not included.

The outline of the modelling set up is depicted in Fig. 1. The random forest model for predicting KIR gene content was implemented with R v4.0.4 (*R Core Team, 2021*) using the library ranger v0.12.1 (*Wright & Ziegler, 2017*). The model error was estimated by dividing the original 818 samples randomly into two approximately equal subsets to one of which a RF model for each KIR gene was fitted while the other subset was used for prediction with the fitted model. SNP dosage values in the KIR region on chr19 were used as predictor variables, and the KIR gene content (1 for presence, 0 for absence) as determined by targeted

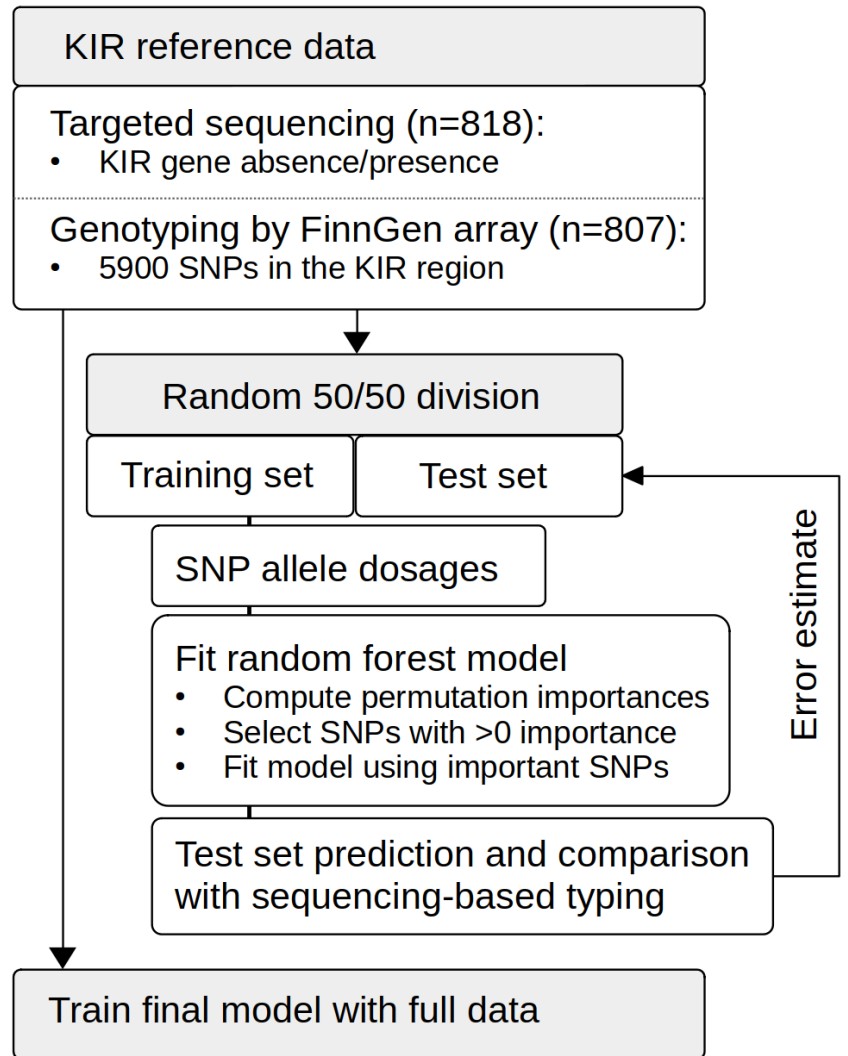

**Figure 1** **Schematic presentation of the study setup.** The reference data set of 818 individuals was genotyped on the FinnGen SNP array, and KIR gene content was determined by targeted sequencing. We used random forests to fit models to the training data set comprising randomly selected 402 individuals. Feature selection was based on the importance metric computed through permutation. The model was refitted on SNPs achieving an importance of < 0. Based on the test set comprising the other half of the samples ($n = 405$), we calculated prediction error estimates for the modeling approach. Finally, we used the whole data set ($n = 807$) to train complete models.

sequencing served as the target phenotype variable. Feature selection within the model fitting was implemented using the permutated importance metric. Variants achieving an importance $> 1 \times 10^{-5}$ were accepted into the model. The final model was fitted with the full dataset (Fig. 1).

Accuracy metrics were calculated using the R library caret v6.0-86 *Kuhn (2020)*. Positive predictive value (PPV) was defined as (sensitivity * prevalence)/((sensitivity*prevalence) + ((1-specificity)*(1-prevalence))). Negative predictive value (NPV) was defined

as (specificity * (1-prevalence))/(((1-sensitivity)*prevalence) + ((specificity)*(1-prevalence))). Balanced accuracy was calculated as (sensitivity+specificity)/2. Overall accuracy was calculated as the proportion of correct calls from all calls, and 95% confidence intervals for the accuracy estimates were determined by binomial distribution. The data were managed with the tidyverse v1.3.0 (*Wickham et al., 2019*) package system.

To compare our method with KIR*IMP, KIR*IMP v1.2.0 (http://imp.science.unimelb.edu.au/kir/) (*Vukcevic et al., 2015*) was applied to the 818 samples constituting our reference panel. Prior to submitting the data to KIR*IMP, the genotypes were transferred to hg19 coordinates with UCSC LiftOver (http://genome.ucsc.edu/cgi-bin/hgLiftOver) and phased with shapeit v2.r904 (*O'Connell et al., 2014*) with default parameters except for burn = 10, prune = 10, main = 50 and window = 0.5. The orientations of the SNPs in the phased dataset were harmonised according to the KIR*IMP SNP information file (http://imp.science.unimelb.edu.au/kir/static/kirimp.uk1.snp.info.csv) using a custom R script. In analysing the results, we used KIR2DS4TOTAL and KIR3DL1ex9 CNV imputation results of KIR*IMP to compare with our KIR2DS4 and KIR3DL1 absence-presence imputations, respectively.

### Code and data availability

The analysis code and the models are available at https://github.com/FRCBS/KIR-imputation https://github.com/FRCBS/KIR-imputation. SNP genotype data have been deposited at the European Genome-phenome Archive (EGA), which is hosted by the EBI and the CRG, under accession number EGAS00001005457. Further information about EGA can be found on https://ega-archive.org "The European Genome-phenome Archive of human data consented for biomedical research" (http://www.nature.com/ng/journal/v47/n7/full/ng.3312.html).

## RESULTS

Accuracy estimates for each KIR gene for prediction of an independent test set are listed in Table 1. In summary, the mean overall accuracy of prediction was 0.985 (95% CI [0.970–0.992]). The lowest accuracy of 0.956 (95% CI [0.937–0.973]) was obtained for KIR2DL5 while KIR2DL2, KIR2DL3, KIR2DS2, KIR2DS4 and KIR3DL1 all achieved an overall accuracy of 1. Accuracy estimates for the test data and the RF out-of-bag (training set and full data) are plotted in Fig. 2A. SNPs used by the models are listed in Table S1.

To compare our approach with KIR*IMP, we converted our dataset of 807 samples to hg19 genome build and harmonised the SNP orientations. 126 SNPs out of 5900 could not be lifted over to hg19, and out of the 301 SNPs used by KIR*IMP 249 were found in our input data. SNP allele frequencies between the KIR*IMP reference panel and the input data had Pearson's correlation coefficient of 0.968 (Fig. S1A). Mean accuracy based on an estimate from the KIR*IMP reference subsetted for the input SNPs was 96.59% (Fig. S1B). Accuracy metrics for the imputation of our data by KIR*IMP are indicated by grey colour in Fig. 2A. In summary, for all 12 included KIR genes we observed a distinctly lower imputation accuracy for KIR*IMP in comparison with our method.

**Table 1  KIR imputation accuracy.**

| KIR_gene | Sensitivity | Specificity | Pos.Pred.Value | Neg.Pred.Value | Precision | Recall | Balanced.Accuracy | Accuracy |
|---|---|---|---|---|---|---|---|---|
| 2DL1 | 0.857 | 0.995 | 0.75 | 0.997 | 0.75 | 0.857 | 0.926 | 0.993 |
| 2DL2 | 1 | 1 | 1 | 1 | 1 | 1 | 1 | 1 |
| 2DL3 | 1 | 1 | 1 | 1 | 1 | 1 | 1 | 1 |
| 2DL5 | 0.948 | 0.962 | 0.958 | 0.953 | 0.958 | 0.948 | 0.955 | 0.956 |
| 2DP1 | 1 | 0.995 | 0.75 | 1 | 0.75 | 1 | 0.997 | 0.995 |
| 2DS1 | 0.986 | 0.989 | 0.991 | 0.984 | 0.991 | 0.986 | 0.988 | 0.988 |
| 2DS2 | 1 | 1 | 1 | 1 | 1 | 1 | 1 | 1 |
| 2DS3 | 0.983 | 0.912 | 0.971 | 0.949 | 0.971 | 0.983 | 0.948 | 0.965 |
| 2DS4 | 1 | 1 | 1 | 1 | 1 | 1 | 1 | 1 |
| 2DS5 | 0.969 | 0.953 | 0.973 | 0.946 | 0.973 | 0.969 | 0.961 | 0.963 |
| 3DL1 | 1 | 1 | 1 | 1 | 1 | 1 | 1 | 1 |
| 3DS1 | 0.968 | 0.957 | 0.964 | 0.962 | 0.964 | 0.968 | 0.962 | 0.963 |

To compare the level of overall accuracy of our SNP-based method with an established sequencing-based approach, we extracted the results of the evaluation by Chen and co-workers (*Chen et al., 2020*) for the KIR imputation method kpi (*Roe & Kuang, 2020*). Fig. 2B shows the accuracy of kpi compared with our test set results. The observed values were highly similar with KIR2DS3 being the most difficult gene to impute correctly.

Varying numbers of missing SNPs within the KIR region reduced the imputation accuracy in accordance with the fraction of removed variants. At 80% of the SNPs present the accuracy generally remained at a good level but started to increasingly deteriorate after that (Fig. 2C).

Imputation posterior probability (PP) values are potentially informative of imputation uncertainty and can be incorporated into association analyses (*Zhou et al., 2020*). Figure 2D shows the PP distributions for each imputed KIR gene. Figure 2E shows confusion tables for KIR gene absence/presence classifications. Here, the presence of each imputed KIR gene was determined by the criterion PP > 0.5; otherwise the gene was determined to be absent.

Figure 3A shows the effect of PP-based filtering on imputation accuracy. More stringent PP thresholds slightly improved accuracy estimates for KIR2DL1, KIR2DP1, KIR2DS3, KIR2DS5 and KIR3DS1. The number of samples discarded as a result of PP filtering varied between genes (Fig. 3B); for about half of the KIR genes the filtering had no effect.

## DISCUSSION

Genome data generated in a growing number of biobank projects is instrumental to detailed immunogenetic analyses of several clinical phenotypes and diseases. Within the current technological and economical constraints SNP microarrays offer a practical way for genotyping hundreds of thousands of individuals. The KIR gene content, despite being a relatively coarse-scale feature, has been shown to influence many immune-mediated disorders (*Bashirova et al., 2006*; *Parham & Guethlein, 2018*) and complications in pregnancy (*Colucci, 2017*). Imputation of KIR gene content from SNPs in a scalable way is therefore essential to analysing and interpreting large biomedical databases. To this

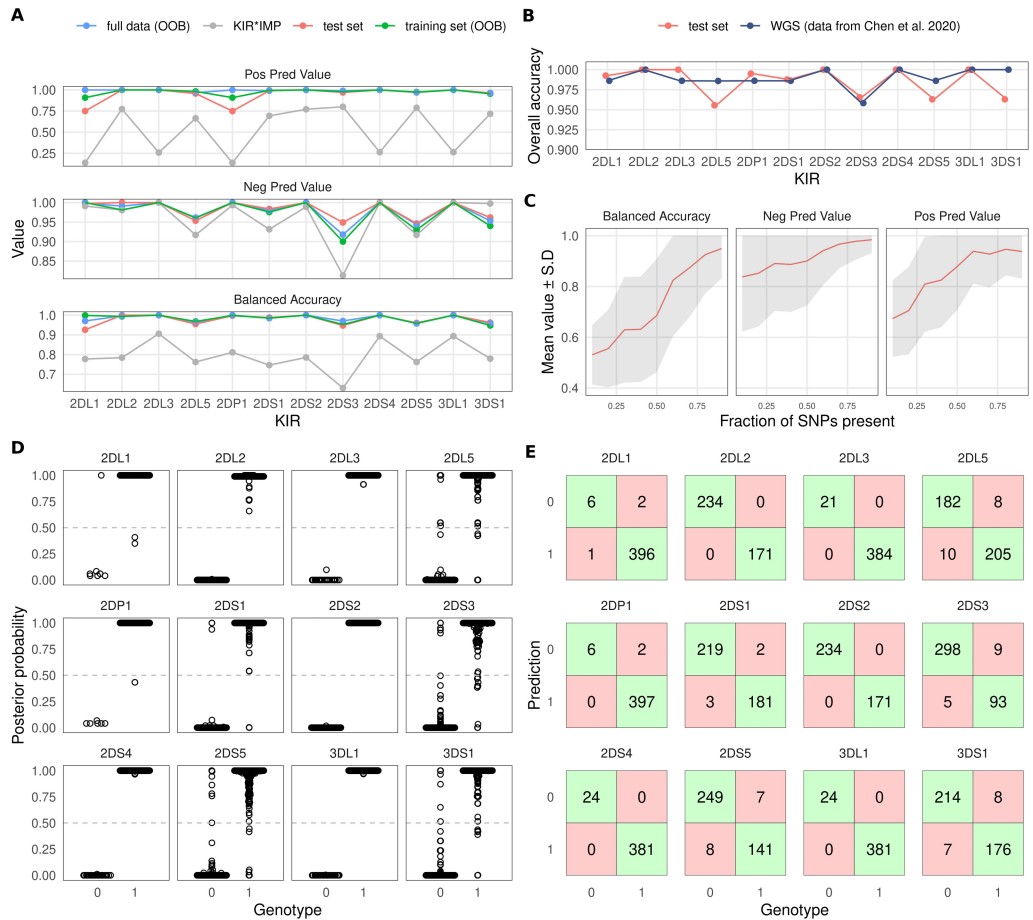

**Figure 2** **Overview of KIR prediction accuracy.** (A) Prediction performance metrics for the 12 imputed KIR genes. OOB: out-of-bag estimate from random forest models. Test set was predicted by models fitted on the training set. KIR*IMP was applied to the full dataset. Note the varying scale of the *y*-axis. (B) Comparison of overall accuracies between the test set and reported values for the WGS based method kpi extracted from the publication by Chen and co-workers. (C) Impact of missing SNPs on prediction performance in the test set. (D) Posterior probability distributions for test set prediction. Genotypes 0 and 1 denote absence and presence of a KIR gene, respectively. (E) Confusion tables for the test set prediction. Posterior probabilities >0.5 were classified as 'present'.

end, in the present study we have built a machine learning model for inferring KIR gene content from SNP dosage data for stand-alone application in biobanks and other clinical data collections. Exploitation of random forest for imputing KIRs from SNP genotypes was first implemented in the KIR*IMP software (*Vukcevic et al., 2015*), which runs on a remote server (http://imp.science.unimelb.edu.au/kir/). The main difference of our method in comparison with KIR*IMP is that it does not require phased data, the models can be downloaded and run locally, and the user can build their own models. However, KIR*IMP produces a more detailed output that includes A and B haplotypes, framework genes KIR3DP1 and KIR2DL4, variants of KIR2DS4 and KIR3DL1 and gene copy numbers. Otherwise, at the level of gene absence-presence, the imputation accuracy of our method

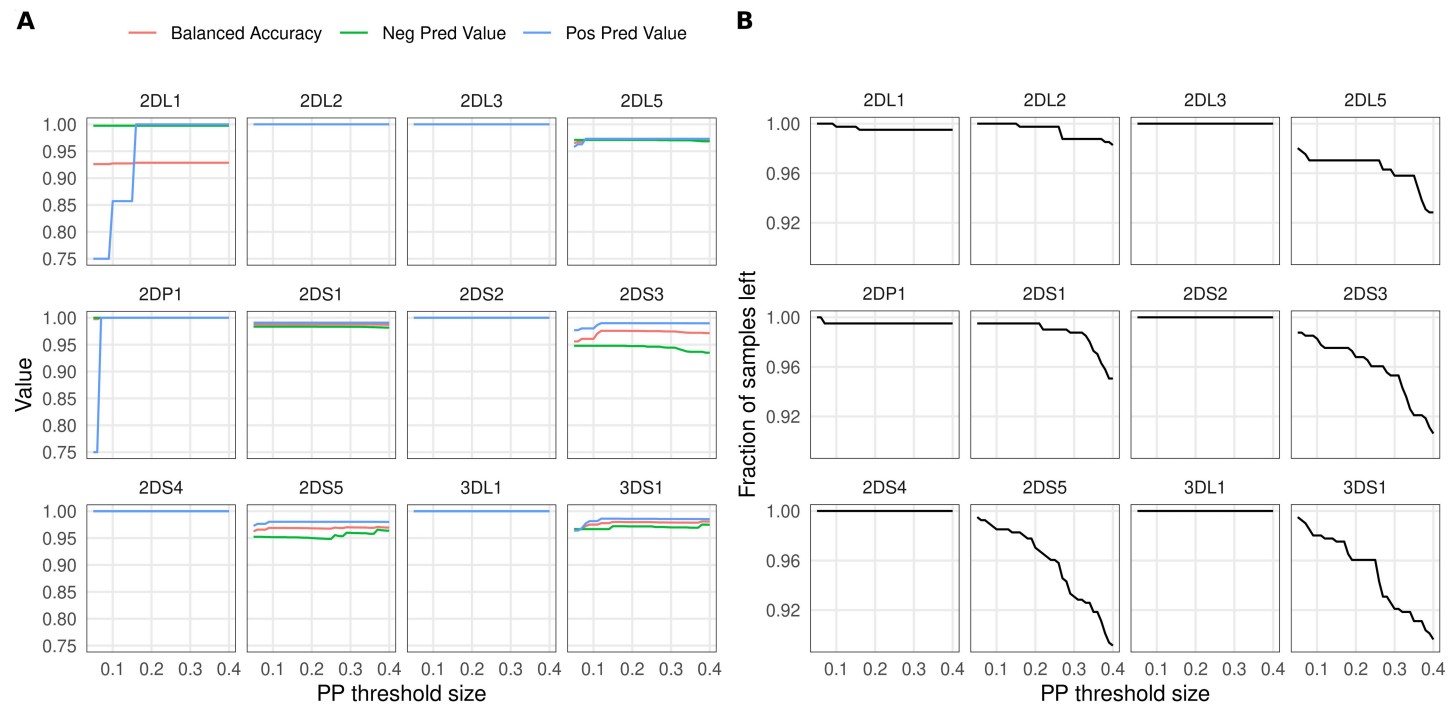

**Figure 3** **Effect of posterior probability on imputation accuracy.** Imputation results were filtered using varying posterior probability thresholds prior to calculating accuracies. The *x*-axis shows the threshold size relative to 0.5: for example, a value of 0.1 sets the lower threshold for KIR gene presence at 0.6 and the upper threshold for gene absence at 0.4. (A) Accuracy estimates (*y*-axis) for the 12 KIR genes at different threshold values. (B) Fraction of samples left (*y*-axis) after applying the thresholds.

compares favourably not only with KIR*IMP but also to sequencing-based methods (*Chen et al., 2020*; *Roe & Kuang, 2020*).

In all imputation evaluations KIR2DS3 demonstrated the largest error in overall accuracy, followed by KIR2DL5 and KIR2DS5. A common feature shared by these three genes is that their location within the KIR chromosomal region is not fixed but can vary between centromeric and telomeric positions (*Hsu et al., 2002*; *Pyo et al., 2010*). Conceivably, this kind of positional variance may confound the identification of predictive SNPs resulting in greater imputation uncertainty. Other challenging genes were KIR2DL1 and KIR2DP1 which both harbour a relatively rare gene absence with population frequency of about 1.6%, and therefore had few cases in the training data. In this regard, the out-of-bag estimate for the whole dataset might be the most reliable error estimate for these genes, suggesting a balanced accuracy and positive predictive value of about 0.95. Despite some challenges, KIR gene content imputation presents a valuable tool for initial screening and provides a rational basis for further analyses.

Allelic diversity and homologous gene sequences make KIR typing challenging by NGS or microarray probes. Moreover, the genetic complexity of the KIR region is also evidenced by segment duplications which can result in haplotypes carrying more than two copies of a KIR gene (*Norman et al., 2009*). This can negatively affect SNP-based imputation accuracy especially in rarer haplotypes that may not be well represented in the model training data.

Nevertheless, the large number of variants within the genomic region allows extraction of information based on linkage with gene content, even if the causative variants cannot in all cases be directly measured. This is also a shortcoming because linkage patterns vary between populations and consequently models trained on one population may not be fully transferrable to another. While the informative SNPs used by our method are not specific to the Finnish population as such, but present a set of common genetic variants with relatively similar allele frequencies across European populations, it is not guaranteed that the prediction would achieve as good an accuracy in populations other than Finns. Therefore, for translating the method to other populations, the best option would be building population-specific models or using a large training data set that captures most of the existing genetic diversity in KIRs.

In estimating the model error we used cross-validation in a randomly selected half of our sample set. The full published model, however, was trained on all samples, and estimating the error of the full model is based solely on out-of-bag (OOB) samples from within the model bootstraps. Thus, we are assuming that the OOB error captures the true model error at a reasonable accuracy. This assumption was supported by our data from the cross-validation where the OOB error from the training set closely followed the estimate from the independent test set. Furthermore, as the full model is trained on all the samples, it can be assumed to better capture the KIR diversity present in the population.

Our method is also limited by the requirement of the availability of informative SNPs in the dataset under analysis. These variants are not genotyped by all microarrays commonly used in genome analysis and therefore selection of a suitable platform is crucial. Another noteworthy limitation is that the method is not capable of identifying alleles. To date, only targeted sequencing-based approaches can resolve KIR alleles (*Norman et al., 2016*; *Maniangou et al., 2017*; *Wagner et al., 2018*; *Roe & Kuang, 2020*).

## CONCLUSIONS

Linkage disequilibrium patterns vary between human populations and potentially affect imputation of complex genotypes including KIR variation. Here we have studied KIR gene content imputation in the Finnish population to build a method suitable for biobank data and to evaluate the advantages of a population-specific reference. The results based on cross-validation demonstrate a good overall accuracy and highlight the importance of matching the imputation reference panel to the target population. As KIR function is modified by considerable allelelic diversity, a possible future direction is to extend KIR imputation from SNPs to cover allelic diversity for more detailed analysis of immunogenetic associations.

## ACKNOWLEDGEMENTS

We thank the Finnish Red Cross Blood Service Biobank and the blood donors for providing the samples. FinnGen is acknowledged for providing the SNP genotype data.

### Funding

This work was supported by the Academy of Finland, the Finnish Cancer Association, VTR funding from the Finnish Government, and Business Finland. The funders had no role in study design, data collection and analysis, decision to publish, or preparation of the manuscript.

### Grant Disclosures

The following grant information was disclosed by the authors:
The Academy of Finland.
Finnish Cancer Association.
VTR funding from the Finnish Government, and Business Finland.

### Competing Interests

The authors declare there are no competing interests.

### Author Contributions

- Jarmo Ritari conceived and designed the experiments, performed the experiments, analyzed the data, prepared figures and/or tables, authored or reviewed drafts of the paper, and approved the final draft.
- Kati Hyvärinen, Jukka Partanen and Satu Koskela conceived and designed the experiments, authored or reviewed drafts of the paper, and approved the final draft.

### Human Ethics

The following information was supplied relating to ethical approvals (i.e., approving body and any reference numbers):

The Finnish Red Cross Blood Service Biobank approval to use the data to perform this study based on the Finnish Biobank Act (688/2012) (IRB reference number 002-2018).

### DNA Deposition

The following information was supplied regarding the deposition of DNA sequences:

The SNP genotype data are available at the European Genome-phenome Archive (EGA): EGAS00001005457.

The data cannot be made publicly available because these data are sensitive, potentially making it possible to identify individuals, so they can only be made available upon application to EGA. The data will be made available to anyone subject to their agreeing to not republish the data or share with a third party. Please apply to the institutional data protection body by email for access: tietosuojavastaava@veripalvelu.fi.

https://ega-archive.org/datasets/EGAD00010002206.

### Data Availability

The primary SNP data are available at the European Genome-phenome Archive (EGA): EGAS00001005457.

The analysis code is available in GitHub: https://github.com/FRCBS/KIR-imputation.

**Supplemental Information**

Supplemental information for this article can be found online at http://dx.doi.org/10.7717/peerj.12692#supplemental-information.

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
