# Peer review of "KIR gene content imputation from single-nucleotide polymorphisms in the Finnish population"

_PeerJ, doi:10.7717/peerj.12692_

## Round 0.1 · original submission · Major Revisions

Please address concerns of all reviewers and revise manuscript accordingly.

·

Basic reporting

The structure and writing follow standard conventions and are easy to understand. The figures are appropriate and support the text. The literature references are also appropriate and sufficient. The supporting data is sufficient and open.

The goals and claims are clear and modest. The appropriate amount of evidence and discussion are provided.

Experimental design

The manuscript describes an investigation to improve Finnish blood bank services by developing an efficient and accurate SNP panel and genotyping workflow. It is well defined, relevant, and meaningful. The claims do not exceed the design. The methods meet or exceed high technical and ethical standards and, with a few small improvements, contain enough information for reproduction.

Validity of the findings

The goal is clear. The methods and data reflect that goal, have been provided, and are appropriately detailed. The conclusions are also in line with the other sections and are appropriate.

Additional comments

I recommend these items be addressed.

a) The github test doesn’t run for me
Rscript ./src/run_KIR_imputation.R "./results/models" "./test/simulated_ref.raw" "./test/simulated_ref.bim" "./test/output"
Error: No models found! Stopping.
I'm running R 4.0.3 on OS X 11.6.
b) The link on line 94-95 needs altering. It looks like there is an erroneous space in the URL.
c) I’m not quite sure about the meaning of the sentence in 117-118. Why have this cutoff? How many genotypes were affected (not counted) because of it? Does this have to do with why the overall cohort size is 818, but the evaluation cohort is less than half of that? It seems to me that all testing individuals/genes should be considered in the accuracy reporting, or the reasons should be explained more.
d) The figures and tables should have “n=” information in them or their captions.
e) The text doesn’t state the 12 genes that were genotyped nor why they were selected. It appears this decision was based on IPD-KIR gene definitions, excluding the frameworks. Is his correct?
f) Did the validation genotyping confirm the assumption that every individual contains every framework gene?

These items are suggestions.

g) The SNPs were from the FinnGen project, but that is the only information provided about how/why these markers were selected. It would be valuable if more information could be written or referenced about how the SNPs for the KIR region were included in the panel. Of special interest is how markers for B haplotype genes were selected.
h) The text does not mention how many SNPs are used by the algorithm. It is 743, I believe. It would aid the text to add this information and break down the SNP counts by gene. Do the gene names in the SNP names reflect the genes they originate from, mark, or both?
i) It would be nice if the github page would describe the expected input, in addition to referencing the publication. Also, the required packages.
j) The submission metadata says there are 3 figures, but I believe there are only 2 (and 1 table).
k) The web page referred to in lines 134-135 displays two studies: EGAS00001005457 and EGAD00010002206?. They look the same. Why are there two, and what are the differences?

Reviewer 2 ·

Basic reporting

Overall the manuscript is well-written. I did have a problem with the paragraph in the introduction encompassing lines 48-60. First, in line 50 the authors state the A haplotype has 'functional' 3DL3, etc genes, but then later in the paragraph goes on to state that many of the 2DS4 genes encode non-funtional receptors. 'Functional' should be removed. Second, the genes listed are neither activating or inhibitory, instead they encode activating or inhibitory receptors. The text should be modified to reflect this. For example, '...harbours several activating KIR genes...' should be changed to '..harbours several genes encoding activating KIR...'.

With regard to references, in line 70 they state that KIR*IMP is the only other known imputation method available, but then in the Results section (line 155) they mention a second method, kpi. Reference to it should be included in the introduction.

Experimental design

The method is well-described and appears highly accurate for the populations in which it was tested. Data, code and models are available. It will be interesting to see how well this method will perform when presented with data from more diverse populations.

Validity of the findings

The authors have properly stated that their training and testing focused on European populations and that adding geographically/ethnically diverse populations may impact the accuracy of their predictive model.

In the discussion of results for 2DL5, 2DS3, and 2DS5, the authors comment on positional variability. Along with that comes the possibility of 0-4 copies in any individual. It would be helpful if the authors could comment on how that may affect the accuracy of their prediction. In the same vein, there are known haplotypes that contain duplication of segments of the KIR region. Are the authors aware of any such haplotypes in their dataset and how that copy number variation (3 copies were two maximum are expected) would affect their results?

·

Basic reporting

The article is well written, and structured in a way that it's straightforward to follow the rationale. The introduction provides sufficient context, and the amount of cited literature is appropriate. Figure are very clear and nicely presented.

Experimental design

The authors describe a novel method to impute KIR gene presence from SNP genotyping data. The study was well designed and executed, and all methods are described with sufficient detail. The authors rightly state that there is only one SNP-based approach for KIR imputation thus far, which requires an upload of individual-level genetic data to an online server. This is often not possible due to genetic privacy regulations, and thus the authors' efforts present an important and welcome contribution to the field of immunogenetics. One possible issue is the exclusion of patients based on probability scores before calculating imputation accuracy (see general comments).

Validity of the findings

The conclusion is well supported by method description and results. The authors clearly state that their method was developed for the Finnish population. However, it seems that the workflow itself can be replicated in other populations relatively easily, and I'm sure others, including yours truly, would appreciate the authors' point of view on such efforts. It would be great to add a paragraph to the discussion explaining what would be required to translate this to populations of different genetic ancestry.

Additional comments

1) Numbers in Figure 2E do not add up to 409, as I would have expected based on Figure 1. How did the author lose 8% of their cohort? The description of how posterior probability was used is not clear to me (Lines 161-166). Did the authors remove individuals according to a certain threshold, before reporting accuracy estimates? If yes, this warrants a clear description and more discussion. One might argue that low PP for a significant fraction of the cohort is indeed a measure of accuracy, and that it's not appropriate to simply disregard that. My preference would be a more conservative approach, first reporting accuracy independent of this filtering, and then explaining that PP can be used to remove likely false results from a cohort one intends to analyze, which in turn leads to very high accuracy for the 92% of individuals that can be considered.

2) Figure 2b might require some clarification, since it appears as if the accuracy estimates for the novel software vs. kpi ("WGS (external)") were from the same evaluated data set, which is not the case. So it is not a direct comparison, but the figure shows independently generated accuracy estimates from different populations.

3) A short description of how the reference data was generated (apart from the name of the vendor) would be appreciated.

·

Basic reporting

no comment

Experimental design

The study is designed in a standard way of other SNP imputation-based papers.
The method is straight forward and comparisons with existing software are clearly proven.
The limitation of the panel is clearly stated

Validity of the findings

Internal validation is performed and high accuracy is achieved, however; the review is concerned about the problem of overfitting and validation with an independent dataset is necessary to validate the accuracy of the reference.

---

## Round 0.2 · Minor Revisions

As you can see, reviewers are mostly satisfied by your revision. However, reviewer #3 has a minor concern requesting your response. Please address the remaining issues and revise manuscript accordingly.

·

Basic reporting

I find the requested changes to be acceptable.

Experimental design

I find the requested changes to be acceptable.

Validity of the findings

I find the requested changes to be acceptable.

Additional comments

I find the requested changes to be acceptable.

Reviewer 2 ·

Basic reporting

The manuscript text did not match the rebuttal text for the requested correction of line 50 (and the remainder of the paragraphJ. I would suggest amending as follows for clarity.

The KIR gene cluster on the human chromosome 19q13.4 encodes fifteen relatively homologous KIR genes and two pseudogenes, constituting two main haplotypes: A and B (https://www.ebi.ac.uk/ipd/kir/sequenced_haplotypes.html). The group A haplotype consists of functional KIR3DL3, KIR2DL3, KIR2DL1, KIR2DL4, KIR3DL1, KIR2DS4 and KIR3DL2 genes of which all except KIR2DS4 encode inhibitory KIR. The group B haplotype, on the other hand, is more diverse being characterised by the presence of at least one of KIR2DS2, KIR2DL2, KIR2DL5, KIR2DS5, KIR3DS1, KIR2DS3 or KIR2DS1 genes (Bashirova et al., 2006). Thus, the group B haplotype harbours several genes encoding activating KIR, whereas the only activating receptor, KIR2DS4, encoded by group A is a non-functional truncated variant in a significant proportion of Caucasians (Maxwell et al., 2002; Bontadini et al., 2006), rendering about 40% of group A homozygotes solely inhibitory. Approximately 55% of haplotypes are mixtures between group A and B (Middleton & Gonzelez, 2010), making the haplotype strucure highly variable in the population. Allelic diversity within KIRs is equally high with at least a few hundred known polymorphisms (https://www.ebi.ac.uk/ipd/kir/stats.html), which can affect class I ligand affinity (Carr, Pando & Parham, 2005; Frazier et al., 2013).

Experimental design

The authors have addressed my concerns.

Validity of the findings

No additional comments.

·

Basic reporting

The revised sections of the text are clear and well written.

Experimental design

no comment

Validity of the findings

no comment

Additional comments

I appreciate the addition of a training script to the repository that will allow training building custom imputation models.

Figure 1 now states that 807 samples were included, and 50% randomly assigned to training and test set. This still doesn't explain the sum of 405 in the confusion tables (Figure 2E). It's a minor discrepancy (I'd expect 403 or 404 based on the 50/50 assignment), but it should be explained.

---

## Round 0.3 · accepted · Accept

All remaining issues of the reviewers were addressed and the amended manuscript is acceptable now.